

# Association of heat shock protein 8 with atopic march in a murine experimental model

Kyu-Tae Jeong, Ji-Hye Do, Sung-Hee Lee, Jeom-Kyu Lee and Woo-Sung Chang

Division of Allergy and Respiratory Disease Research, Department of Chronic Disease Convergence Research, Korea National Institute of Health, Korea Disease Control and Prevention Agency, Cheongju, Osong-eup, Heungdeok-gu, Korea

## ABSTRACT

**Background:** Atopic march (AM), a unique characteristic of allergic diseases, refers to the sequential progression of atopic dermatitis (AD) in infants to allergic asthma and allergic rhinitis in children and young adults, respectively. Although there are several studies on AM, the establishment of an AM murine model to expand our understanding of the underlying mechanism and to identify the potential biomarkers is yet to be achieved. In this study, an improved murine model was established by applying a method to minimize skin irritation in inducing AD, and it was used to perform integrated analyses to discover candidate biomarkers.

**Methods:** To induce atopic dermatitis, 2,4-dinitrochlorobenzene (DNCB) was applied to the ear skin once a week, and this was continued for 5 weeks. From the second application of DNCB, *Dermatophagoides pteronyssinus* (Dp) extract was applied topically 2 days after each DNCB application; this was continued for 4 weeks. Dp sensitization and intranasal challenges were then performed for 4 weeks to develop conditions mimicking AM.

**Results:** Exacerbated airway inflammation and allergic responses observed in the AM-induced group suggested successful AM development in our model. Two-dimensional gel electrophoresis (2-DE) and mass spectrometry analysis identified 753 candidate proteins from 124 2-DE spots differentially expressed among the experimental groups. Functional analyses, such as Gene Ontology (GO) annotation and protein–protein interaction (PPI) analysis were conducted to investigate the relationship among the candidate proteins. Seventy-two GO terms were significant between the two groups; heat shock protein 8 (Hspa8) was found to be included in six of the top 10 GO terms. Hspa8 scored high on the PPI parameters as well.

**Conclusion:** We established an improved murine model for AM and proposed Hspa8 as a candidate biomarker for AM.

Corresponding author
Woo-Sung Chang, cws99@korea.kr

## INTRODUCTION

Over the years, the prevalence and burden of allergic diseases, such as asthma, allergic rhinitis, and atopic dermatitis (AD), have been increasing worldwide (*Eder, Ege & von Mutius, 2006*; *Odhiambo et al., 2009*; *Platts-Mills, 2015*). Atopic march (AM), a distinctive feature underlying allergic disease, is characterized by sequential progression of allergic diseases, such as AD in infants, followed by allergic asthma and allergic rhinitis in children and young adults, respectively (*Aw et al., 2020*; *Cohn, Elias & Chupp, 2004*; *del Giudice, Rocco & Capristo, 2006*). The concept of AM primarily revolves around the fact that the presence of one allergic disease leads to an increased risk for others, suggesting the presence of a causal relationship among allergic diseases (*Bantz, Zhu & Zheng, 2014*; *Hill & Spergel, 2018*). Recently, our understanding of AM has been expanded by several cohort studies and evidence obtained from experimental murine models. Previous studies have revealed the presence of asthma, allergic rhinitis, and one or more atopic comorbidities in infants with greater AD severity (*Gustafsson, Sjöberg & Foucard, 2000*; *Schneider et al., 2016*). A prospective birth cohort study revealed that children with a combination of AM and allergic sensitization in early life are likely to have an increased risk of asthma and food allergies at the age of 3 years (*Tran et al., 2018*). In terms of AM pathogenesis, multiple data from animal models support the hypothesis that exposure to allergens through inflamed skin is the primary route for systemic type 2 inflammation that leads to AM (*Hill & Spergel, 2018*; *Hogan, Peele & Wilson, 2012*).

The concept of AM is considered important and helpful in the early recognition of subsequent diseases and identification of infants at high risk of allergic progression (*Busse, 2018*). Despite a variety of studies on the mechanism or interventions for AM, diverse approaches are warranted to expand the understanding of the relationships between allergic diseases and to develop strategies for preventive interventions. In this regard, a murine model for AM is required to provide insight into the mechanism of AM and find the potential biomarkers that could be utilized in the strategies for AM.

In this study, we aimed to establish a murine experimental model for AM by sequentially provoking asthma after the induction of AD by minimizing skin irritation caused by hair removal. We then applied two-dimensional gel electrophoresis (2-DE) analysis and mass spectrometry (MS) to identify the differentially expressed proteins in the bronchoalveolar lavage fluid (BALF) between control and AM-induced mice. Functional and network analysis were conducted to find candidate biomarkers in AM by investigating the significance of the identified proteins and their interactions.

## MATERIALS AND METHODS

### Animals

Female BALB/c mice (5 weeks old) were purchased from Orient Bio (Seongnam, Korea). The mice were housed in the animal research center of Korea Disease Control and Prevention Agency at a controlled ambient temperature of 22 °C with 50 ± 20% relative humidity under a 12 h light-dark cycle (lights on at 7:00 AM). A total of 60 mice, 20 in three independent sets, were used for this study. We randomly divided the mice into each

experimental group. Animal care and experimental protocols were approved by the Institutional Animal Care and Use Committee of the Korea Centers for Disease Control and Prevention (KCDC-031-16-2A, KCDC-033-17-2A, KCDC-121-17-2A, KCDC-019-19-2A, KCDC-034-20-2A).

## Murine model for AM

The extract of *Dermatophagoides pteronyssinus* (Dp), a major species of house dust mite, purchased from Prolagen (PEA-DERP010; Yonsei University College of Medicine, Seoul, Korea), was re-suspended in phosphate-buffered saline (PBS). To induce AD, a previously published protocol (*Choi & Kim, 2014*; *Kim et al., 2013*) with minor modifications was used. Once a week, 1% 2,4-dinitrochlorobenzene (DNCB) (20 µL of a 4:1 mixture of acetone/olive oil) was applied to the ear skin, and this was continued for 5 weeks (days −7, −2, 5, 12, 19). From the second application of DNCB, 75 µg of Dp (in 20 µL of PBS) or PBS was topically applied 2 days after each DNCB application; this was continued for 4 weeks (days 0, 7, 14, 21). Barrier disruption was achieved by applying 20 µL of 4% sodium dodecyl sulfate to the ear skin 4 h before the application of Dp or PBS. The condition of the skin lesion was measured using digital photographs taken after anesthesia once a week, and plasma was obtained to measure the level of immunoglobulin E (IgE) on days 8 and 22. For the subsequent development of asthma, 4 µg of Dp (in 200 µL of PBS) or PBS was administered intraperitoneally on days 25 and 39, followed by intranasal challenges with 8 µg of Dp (in 40 µL PBS) or PBS for four consecutive days from day 46 to 49 (Fig. 1). Twenty-four hours after the last challenge, the mice were anesthetized and measured airway hyperresponsiveness (AHR) as described below. Immediately following the measurement of AHR, mice were euthanized with an overdose of sodium pentobarbital and samples of plasma, BALF, and lung tissues were obtained for further analysis. PBS-treated mice both in the step of AD induction and asthma development served as the normal control (NC) group. For the AD-induced group, the mice were treated with Dp in the step of AD induction and PBS in the step of asthma development. Mice treated with PBS in the AD step and Dp during asthma development served as the asthma (AS)-induced group. Dp-treated mice in both steps served as the AM-induced group. We established the humane endpoints that the mice would be euthanized under deep anesthesia if any signs, such as weight loss, lethargy, or dyspnea, were observed during all the experiments; however, these were not needed in this study.

## Measurement of AHR

AHR was measured using the flexi-Vent system (flexiVent Fx1; SCIREQ, Montreal, Quebec, Canada) according to the manufacturer's protocol. Briefly, the mice were anesthetized with 50 mg/kg sodium pentobarbital, and tracheostomy was performed. The mice were then intubated using a blunt needle and connected to a small-animal ventilator with a computer-controlled piston. A cycle that included regular ventilation, deep inflation, and exposure to one concentration of methacholine, was performed four times per mouse for about 5 min according to increasing doses of methacholine (0, 12.5, 25, and 50 mg/mL; Sigma-Aldrich, St. Louis, MO, USA) to collect AHR data. This

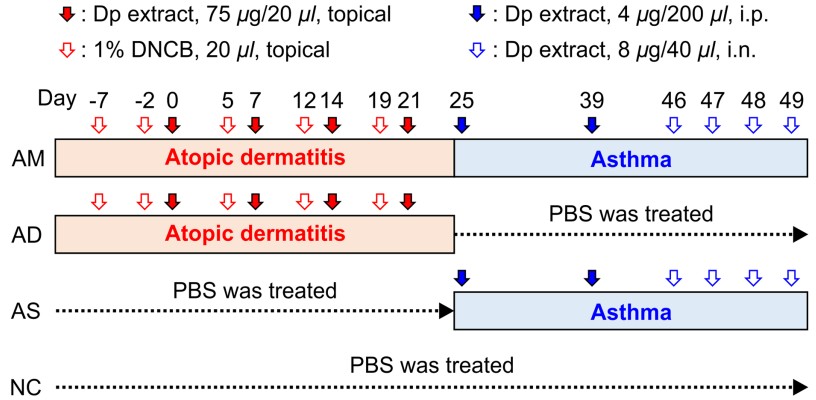

**Figure 1 Experimental protocol for the allergic march model in mice.** The induction of atopic dermatitis (AD) was achieved by the topical application of 1% 2,4-dinitrochlorobenzene (20 μL), followed by 75 μg of *Dermatophagoides pteronyssinus* (Dp) on the ear skin. To develop a barrier disruption, 20 μL of 4% sodium dodecyl sulfate was applied 4 h before Dp application. Allergic asthma (AS) was subsequently induced by two intraperitoneal Dp sensitization and intranasal challenges for four consecutive days. After the last intranasal administration, airway hyperresponsiveness was measured, and then, bronchoalveolar lavage fluid and lung tissue were obtained for further analysis. PBS-treated mice in both the AD induction and asthma development steps served as the normal control (NC) group. The group treated with PBS instead of Dp for each step of AS or AD was designated as AD-only or AS-only induced group, alternatively. Mice were randomly divided into four groups (*n* = 5 per group). 🖼

measurement was conducted by randomly selecting each mouse one by one in the order of NC, AD-induced, AS-induced, and AM-induced groups.

## Enzyme-linked immunosorbent assay

Blood samples, collected and stored for 2 h at ambient temperature of 22 °C, were subjected to centrifugation (4,000 rpm for 10 min), and the supernatants obtained were harvested and stored at −70 °C. Total IgE was measured using a sandwich enzyme-linked immunosorbent assay (ELISA) kit (Biolegend, San Diego, CA, USA) according to the manufacturer's protocol. BALF from each group was collected and centrifuged immediately (13,000 rpm, 5 min). The supernatants obtained were harvested and stored at −70 °C. The levels of interleukin (IL)-4, IL-5, IL-13, and interferon (IFN)-γ in the BALF samples were measured using sandwich ELISA kit (R&D Systems, Inc., Minneapolis, MN, USA), according to the manufacturer's protocol.

## Quantification of Dp-specific IgE

To detect Dp-specific IgE, the antigen-capture ELISA method was used with minor modifications. Briefly, 96-well plates were coated with 10 μg of Dp in 100 μL of coating buffer. After overnight incubation at 4 °C, the plates were blocked with 200 μL/well of assay diluent. Thereafter, 100-μL aliquots of undiluted plasma were added to each well and incubated at room temperature for 1 h. Afterwards, 100 μL of biotin-anti-mouse IgE (BioLegend, San Diego, CA, USA) was added to each well and incubated for 2 h. After incubation with avidin horse radish peroxidase (BioLegend, San Diego, CA, USA) for 30 min, 3,3′,5,5′-tetramethylbenzidine substrate solution (100 μL; Invitrogen, Waltham,

MA, USA) was added to each well and incubated in the dark for 20 min. The reaction was stopped with 2 N sulfuric acid. Optical densities were read at 450 nm with a reference wavelength of 570 nm using the SpectraMax i3x microplate reader (Molecular Devices, San Jose, CA, USA).

## Analysis of immune cells in the BALF

Red blood cells (RBCs) in the precipitated cells obtained from the BALF samples as described above were removed using RBC Lysis buffer (Sigma-Aldrich, St. Louis, MO, USA). The total cells were counted using Nucleo Counter (ChemoMetec, Allerød, Denmark), and 10,000 cells from each sample were spun onto glass slides by cytocentrifugation (Cellspin, Hanil, Kimpo, South Korea) and stained with Diff-Quick solution (Sysmex Corporation, Hyogo, Japan). The number of eosinophils, macrophages, monocytes, lymphocytes, and neutrophils was determined by counting at least 200 cells in each of four different locations of each slide using a microscope (AXIO Imager 2; Carl Zeiss, Oberkochen, Germany).

## Histological analysis of lung tissue

Lobes of the left lung were removed, washed in PBS, and fixed in 4% buffered formalin solution for 3 days. The fixed lung tissues were dehydrated, clarified, and embedded in paraffin. Lung sectioning, subsequent staining with hematoxylin and eosin (H&E; Sigma-Aldrich, St. Louis, MO, USA) and slide scanning were conducted to evaluate general morphology under light microscopy (AXIO Imager 2; Carl Zeiss, Oberkochen, Germany). Lung inflammation was graded using a semiquantitative scoring system as previously described (*Park et al., 2020*). Briefly, peribronchial cell counts were performed blindly based on a five-point scoring system: 0, no cell; 1, a few cells; 2, a ring of cells 1 cell layer deep; 3, a ring of cells 2–4 cells deep; and 4, a ring of cells >4 cells deep. Scoring of inflammatory cells was performed in each lung section for the individual groups, and the mean scores were obtained from three mice that were randomly selected from each group (Supplemental Figure).

## Identification of proteins in the BALF using 2-DE

Proteins in the BALF were precipitated using acetone. Briefly, each BALF sample was added to cold acetone (−20 °C) and incubated for 1 h. After centrifugation (4,000 rpm, 15 min), the pellets were obtained and washed with cold acetone. The pellets were slightly dried and rehydrated with rehydration/sample buffer (8 M urea, 2% CHAPS, 50 mM dithiothreitol (DTT), 0.2% (w/v) Bio-Lyte 3/10 ampholytes, and Bromophenol Blue). After quantification with Bradford assay, the precipitated proteins were separated by 2-DE. The gel was scanned using the ChemiDoc gel imaging system (Bio-Rad Laboratories, Inc., Hercules, CA, USA) to detect the density and distribution of the protein spots. Proteins in excised gel spots were identified with technical support from Proteinworks (Daejeon, Korea) using liquid chromatography-MS/MS (LC-MS/MS) analysis and MASCOT search.

## Functional annotation

The National Center for Biotechnology Information Reference Sequence or Genebank IDs of 2-DE spot proteins were converted to a UniProt Knowledgebase (UniProt KB) IDs using the Database for Annotation, Visualization and Integrated Discovery (DAVID) gene ID conversion tool (*Huang et al., 2008*). The proteins contained actin or albumin and duplicated proteins were eliminated. Next, the functions of candidate proteins were analyzed using DAVID version 6.8 (https://david.ncifcrf.gov/) (*Huang, Sherman & Lempicki, 2009*), which is a web-based functional annotation tool for investigators to analyze the biological roles of genes and is applied to perform Gene Ontology (GO) analysis. For significant GO terms, $p < 0.05$ was considered as the cut-off criterion.

## Protein–protein interaction (PPI) network

A PPI network of proteins was constructed using the STRING database version 11 (http://string-db.org/) (*Szklarczyk et al., 2019*), and the protein interaction relationship network was visualized using Cytoscape software (*Shannon et al., 2003*). The default parameter for selecting a significant interaction pair from the STRING database was 0.4. Furthermore, according to the interaction scores of the PPI network, the Cytoscape plug-in NetworkAnalyzer was used for further analysis. The topological properties of the PPI network and node degree were calculated to search for hub genes from the PPI network. Several different centralities, such as degree, betweenness, closeness, eigenvector, and stress distributions were provided for more screening, but the main connected component of the PPI network was layout by degree values. Degree centrality counts the number of edges at each node and betweenness centrality determines which nodes are important in the flow of the network.

## Statistical analysis

The values are presented as means ± standard error of the mean. Statistical comparisons between groups were conducted using one-way ANOVA and Tukey's test with $p < 0.05$ as the cut-off criterion for statistical significance.

# RESULTS

## Induction of AD-like skin lesions by repeated topical application of Dp

As the first step for the establishment of the AM model, AD was induced by topical application of Dp and DNCB. Afterwards, DNCB was applied once a week for five consecutive weeks, and Dp was applied every 2 days after each DNCB treatment, starting with the second DNCB; this was continued for 4 weeks. We applied them on the ears of the mice to minimize the skin irritation caused by hair removal. AD-like lesions consisting of erythema and excoriation and damage to the epidermal layer had developed in the AD-induced group (Fig. 2A). The total IgE level in plasma was significantly higher in the AD-induced group than in the NC group (Fig. 2B), indicating the induction of the allergic response. These results suggested that AD was well-developed by the repeated topical application of Dp with DNCB only on the ears.

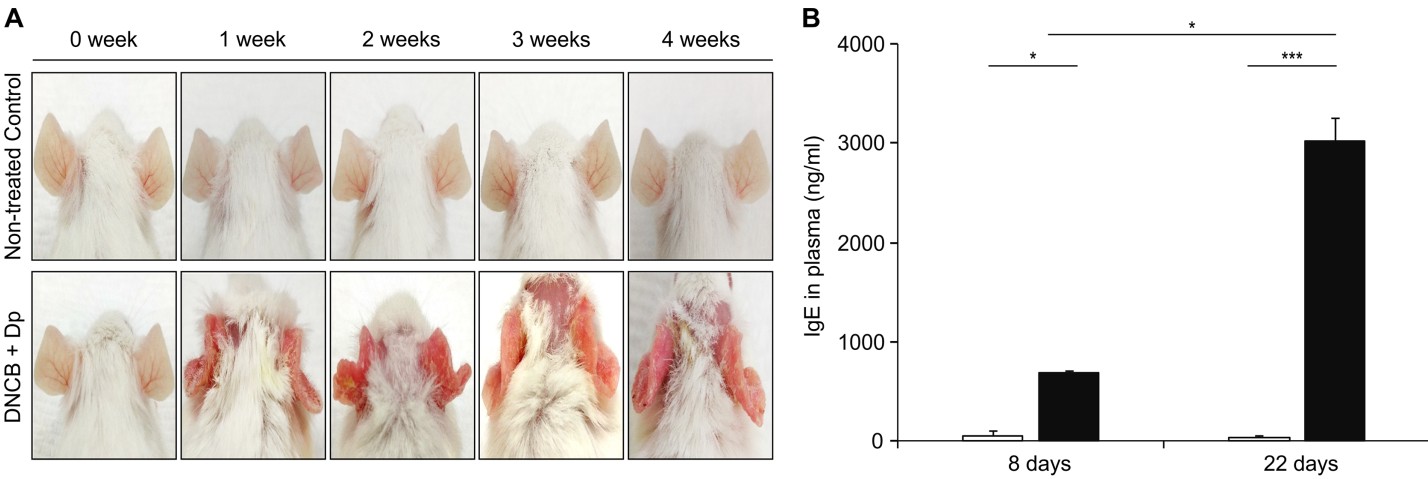

**Figure 2 Induction of atopic dermatitis (AD).** (A) Typical photographs of mouse ears from each group. AD-like lesions were observed at various stages as indicated after starting AD induction by repeatedly applying 2,4-dinitrochlorobenzene and *Dermatophagoides pteronyssinus* (Dp) extract for 4 weeks. Mice in the normal control group were applied PBS instead of Dp. (B) Total immunoglobulin E (IgE) levels in plasma. Blood samples were collected from Dp-applied and normal control mice at day 8 and 22 during the AD induction. The levels of total IgE in plasma was significantly higher in the Dp-applied group than in the normal control group. All data are representative of three independent experiments with similar results. Data are presented as the mean ± standard error of the mean (SEM) ($n$ = 5 per group). $^{*}p < 0.05$ and $^{***}p < 0.001$.

## Exacerbation of AHR and airway inflammation in the AM-induced group

After the successful induction of AD, asthma was sequentially provoked by Dp sensitization and intranasal challenges to develop conditions that mimic AM. We performed Dp challenges daily for the last 4 days, followed by measuring the main parameters that indicate the development of asthma. The values of airway resistance, as a parameter for AHR, were significantly increased in the AM-induced group and asthma-only induced group during inhalation of increasing concentrations of methacholine. The values of airway compliance were significantly decreased in the AM-induced group and asthma-only induced group during inhalation of 25 and 50 mg/mL of methacholine. Interestingly, airway resistance value in the AM-induced group were significantly higher at 50 mg/mL concentration of methacholine than that in the asthma-only induced group (Fig. 3A and 3B). The number of eosinophils in BALF was higher in the AM-induced group than in the other three groups, whereas macrophages/monocytes showed lower level in the AM-induced and asthma-only induced group than in the NC group (Figs. 3C–3G). Cell infiltration observed by H&E staining in lung tissue was also worse in the AM-induced group than in the other groups (Fig. 3H and Supplemental Figure). These results demonstrate that airway inflammation was exacerbated in the AM-induced group that AD and asthma were sequentially induced.

## Highly elevated IgEs and cytokines in the AM-induced group

To investigate whether allergic responses were aggravated under the AM-mimicking conditions, we measured the level of plasma IgEs and various BALF cytokines related to T helper type 2 (Th2) response and inflammation. The levels of total and Dp-specific IgE were significantly higher in the AM-induced group than in the AD or asthma-only induced

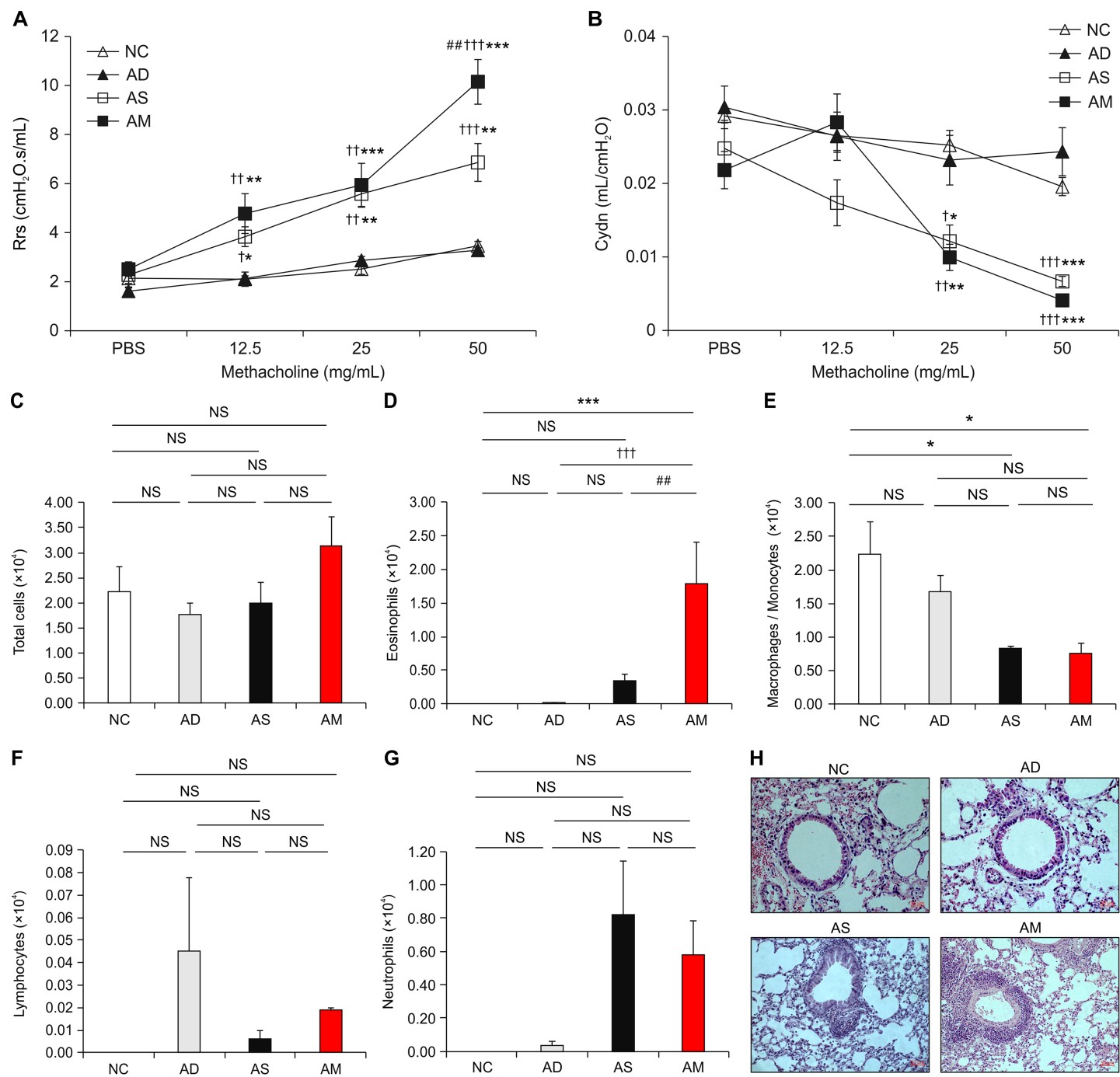

**Figure 3 Analysis of airway hyperresponsiveness (AHR) and airway inflammation in the mouse model of *Dermatophagoides pteronyssinus* (Dp) extract-induced atopic march (AM).** (A and B) Aggravated AHR in response to methacholine. After 24 h of the final intranasal challenge, mice were stimulated with increasing doses of aerosolized methacholine (12.5, 25, and 50 mg/mL). Airway resistance and dynamic compliance were significantly aggravated owing to the development of AM. (C–G) Total cells, eosinophils, macrophages/monocytes, lymphocytes, and neutrophils counts were measured in the bronchoalveolar lavage fluid (BALF) from each group. The number of eosinophils was higher in the AM-induced group than in the other three groups. (H) Representative hematoxylin and eosin-stained sections for lung histology in each experimental group (magnification, 200×; scale bar = 50 μm). Cell infiltration in the lung was worse in the AM-induced group than in the other groups. All data are representative of three independent experiments with similar results. Data are presented as the mean ± standard error of the mean (SEM) ($n$ = 5 per group). Values represent mean ± SEM. $^*p < 0.05$, $^{**}p < 0.01$, and $^{***}p < 0.001$ *vs* the NC group; $^{†}p < 0.05$, $^{††}p < 0.01$, and $^{†††}p < 0.001$ *vs* the AD group; $^{##}p < 0.01$ *vs* the AS group.

groups (Fig. 4A and 4B). Likewise, the Th2 cytokine levels, including those of IL-4, IL-5, and IL-13, were also higher in the BALF obtained from the AM-induced group than that from the other groups (Fig. 4C–4E). However, the level of IFN-γ, a key cytokine for the Th1 response, was significantly lower in the AM-induced group than in the NC and AD-only induced group (Fig. 4F). Another Th1 cytokines, IL-12, was also found to be slightly lower in the AM-induced group compared to the AS-only induced group; however, there was no statistical significance. Thus, the aggravation of the allergic reactions was mediated by Th2 responses under our AM-mimicking conditions. These findings and the exacerbated airway inflammation described above indicate the successful establishment of the murine model for AM.

## Identification of hub proteins and pathways by functional analyses

To identify candidate biomarkers for AM, we investigated the proteins that were differentially expressed among the groups in our model. Through the 2-DE analysis of 28 sets of 92 gels that were loaded with BALF samples from each experimental group, 124 differentially expressed spots (>1.5 fold) were detected. LC-MS/MS analysis and MASCOT search were performed to identify the candidate proteins that each spot represented. A total of 753 candidate proteins, including 406 proteins that were differentially expressed in the AM-induced group compared to the NC, were identified. We then analyzed functional annotation and PPI to determine the biological relationship among the identified proteins. Of the 232 GO terms significantly enriched by GO annotation ($p < 0.05$), 72 GO terms showed a significant enrichment between the AM-induced and the NC groups. As shown in Table 1, six of the Top 10 GO enriched terms were classified into the cellular component group, and we found that Hspa8 was included in all six terms. This was followed by PPI analysis using the STRING database and Cytoscape tool in between the AM-induced and the NC groups. As a result, it also showed Hspa8 as one of the highest-scoring proteins based on the PPI parameters, such as degree, betweenness centrality, and closeness centrality (Table 2); furthermore, Hspa8 was represented as a hub node in the network of the differentially expressed protein (Fig. 5).

## DISCUSSION

In this study, we established a murine experimental model for AM by applying a method for the minimization of skin irritation during AD development. AD-like lesions, such as erythema and excoriation, with elevated total IgE in plasma were observed by the repeated topical application of Dp and DNCB. After the subsequent development of asthma, airway inflammation and allergic responses were aggravated in the AM-induced group, indicating that AM-mimicking conditions were well triggered in our model. Under the concept of AM, the AD development and allergen sensitization in infants predispose them to subsequent development of other allergic diseases, including asthma (*Han, Roan & Ziegler, 2017*). Murine experimental models facilitate the understanding of the underlying mechanisms of AM development and aid in the design of various therapeutic approaches for the prevention and treatment of allergic diseases, in spite of debates citing the inappropriateness of the approach owing to poor reproducibility and

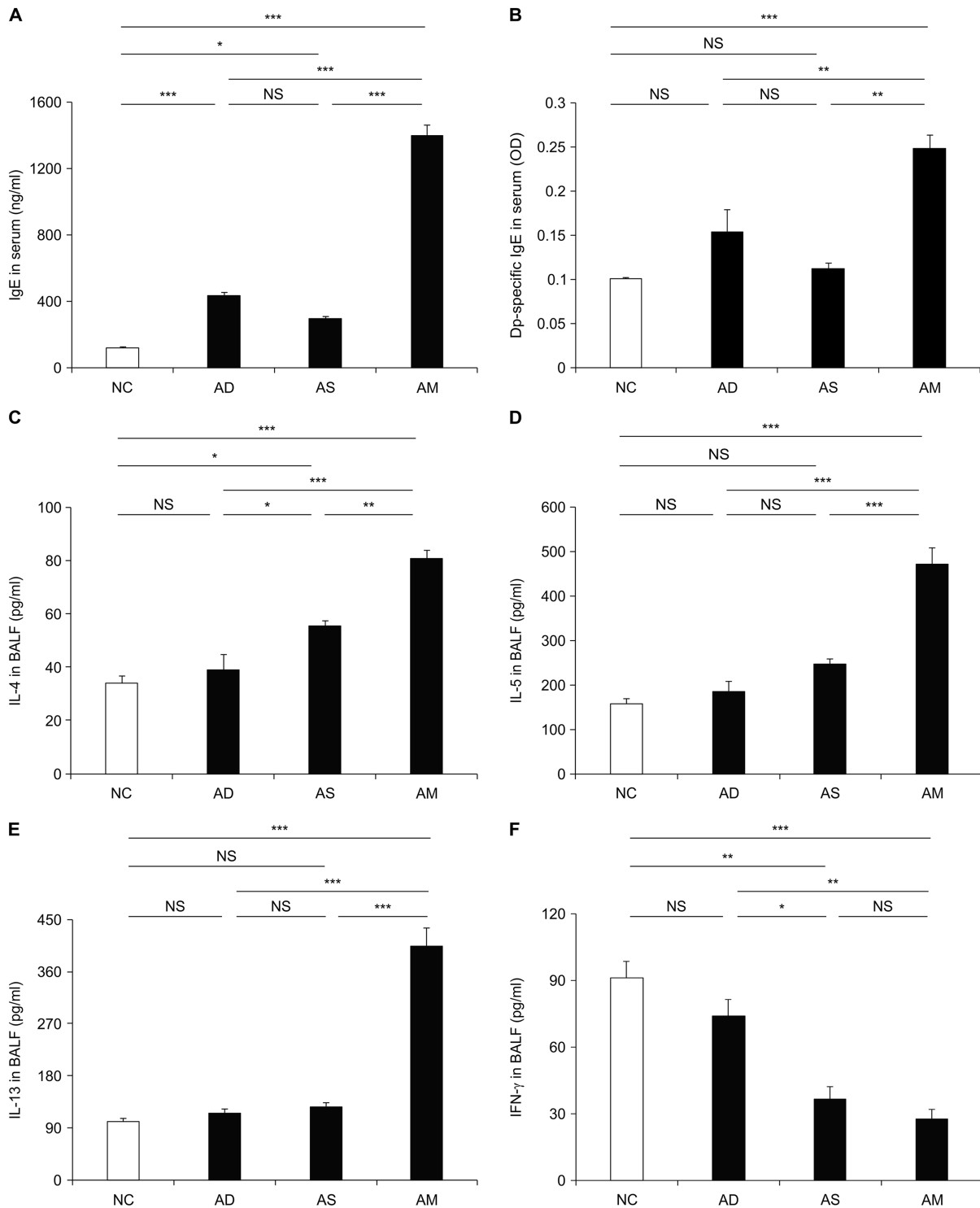

**Figure 4 Elevated level of immunoglobulin E (IgE) and cytokines caused by the development of atopic march (AM).** The plasma levels of total IgE (A) and *Dermatophagoides pteronyssinus*-specific IgE (B) and the levels of interleukin (IL)-4 (C), IL-5 (D), IL-13 (E), and interferon (IFN)-γ (F) in bronchoalveolar lavage fluid were measured using enzyme-linked immunosorbent assay. These data showed a significant increase in allergic responses with the development of AM compared to the other groups. All data are representative of three independent experiments with similar results. Data are presented as the mean ± standard error of the mean (SEM) (*n* = 5 per group). Values represent mean ± SEM. *$p < 0.05$, **$p < 0.01$, and ***$p < 0.001$.

**Table 1 Top 10 GO enrichment terms for differentially expressed proteins between the AM-induced group and the normal control.**

| Category* | Description | Count | P value | Top 3 proteins (total number of proteins) |
|---|---|---|---|---|
| CC | GO:0005615 ~extracellular space | 36 | 9.58E−09 | Heat shock protein 8 Lactotransferrin Collagen, type III, α1 (140) |
| BP | GO:0034097 ~response to cytokine | 9 | 3.23E−07 | Collagen, type III, α1 Serpina1b protein Serpina3f protein (34) |
| CC | GO:0072562 ~blood microparticle | 10 | 1.36E−06 | Heat shock protein 8 Serotransferrin α-fetoprotein (62) |
| CC | GO:0070062 ~extracellular exosome | 45 | 2.11E−06 | Heat shock protein 8 Lactotransferrin Kalirin (160) |
| BP | GO:0043434 ~response to peptide hormone | 8 | 2.21E−06 | Serpina1b protein Serpina3f protein serine peptidase inhibitor (34) |
| CC | GO:0031012 ~extracellular matrix | 12 | 2.82E−05 | Heat shock protein 8 Collagen, type III, α1 Peroxidasin homolog (55) |
| MF | GO:0004867 ~serine-type endopeptidase inhibitor activity | 8 | 8.21E−05 | Serpina1b protein Serpina3f protein serine peptidase inhibitor (17) |
| CC | GO:0005829 ~cytosol | 30 | 2.12E−04 | Heat shock protein 8 Kinesin 2, isoform Peroxiredoxin-6 (139) |
| CC | GO:0005737 ~cytoplasm | 76 | 2.92E−04 | Heat shock protein 8 Kinesin 2, isoform Lactotransferrin (265) |
| BP | GO:0010466 ~negative regulation of peptidase activity | 7 | 4.03E−04 | Serpina1b protein Serpina3f protein serine peptidase inhibitor (16) |

**Note:**
* BP, biological process; CC, cell component; MF, molecular function.

limited translation in humans (*Justice & Dhillon, 2016*). *Lee et al. (2014)* suggested that the repeated application of topical acidic cream in a murine model of AM with oxazolone-induced AD inhibits respiratory allergic inflammation and AD-like skin lesions, suggesting acidification of the stratum corneum (SC) to be a novel intervention method for AM. In another study involving an Nc/Nga mouse-based AM model, they reported the importance of preventing a neutral environment on the SC to alleviate AM-related
**Table 2 Top 10 proteins according to degree, betweenness centrality, and closeness centrality as scored *via* PPI analysis.**

| Proteins | Degree | Proteins | Betweenness centrality | Proteins | Closeness centrality |
|---|---|---|---|---|---|
| Hspa8 | 16 | Rps27a | 0.260756 | Hspa8 | 0.470899 |
| Hspa1b | 14 | Hspa8 | 0.208497 | Hspa1b | 0.451777 |
| Rps27a | 14 | Hspa1b | 0.13845 | Hspb1 | 0.438424 |
| Apoa1 | 11 | Hspb1 | 0.079339 | Rps27a | 0.410138 |
| Hspb1 | 11 | Klc1 | 0.07388 | Trf | 0.402715 |
| Tpi1 | 11 | Vav1 | 0.069638 | Tpi1 | 0.400901 |
| Trf | 10 | Xrcc6 | 0.068084 | Htt | 0.39207 |
| Serpina3n | 10 | Eif3c | 0.067765 | Atp5b | 0.37395 |
| Serpina3f | 10 | Htt | 0.061455 | Anxa5 | 0.369295 |
| Serpina1d | 10 | Col1a1 | 0.055232 | Apoa1 | 0.366255 |

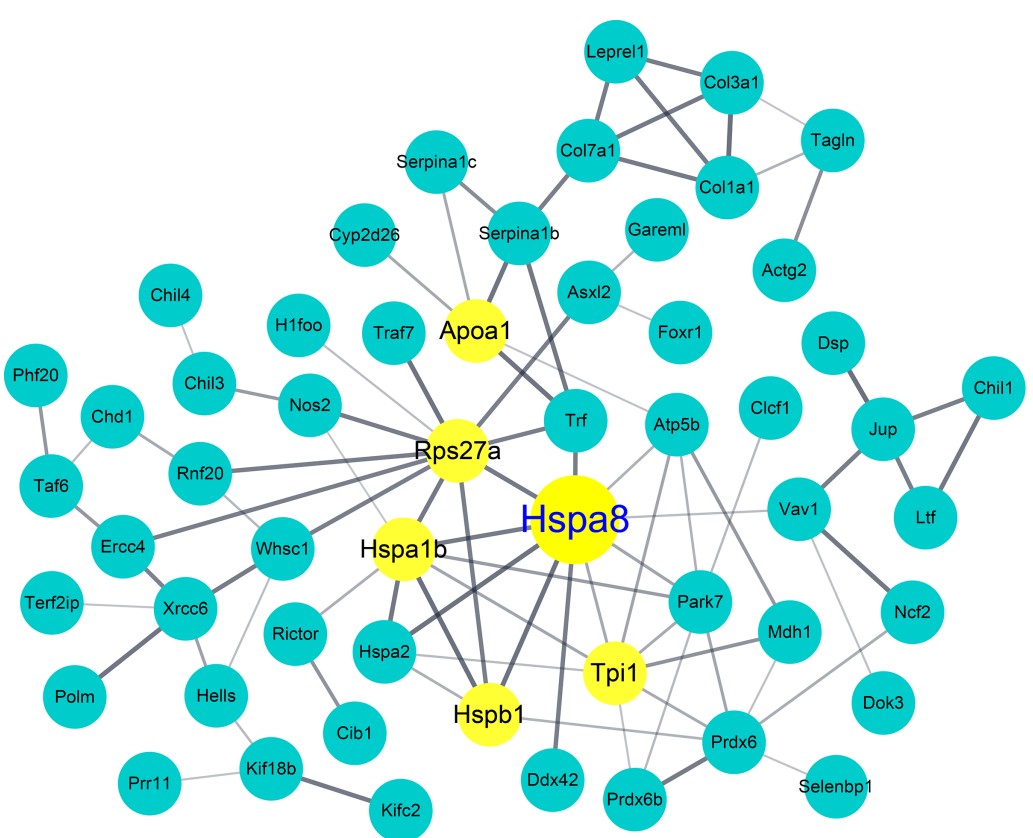

**Figure 5 Protein–protein interaction (PPI) network and identification of heat shock protein 8 as a hub protein.** PPI analysis of differentially expressed proteins in the Atopic march-induced and normal control groups. The yellow node represents hub proteins (degree >10 as cut-off criterion), and the edge represents the interaction relationship among the proteins.

symptoms (*Lee et al., 2015*). Probiotic treatment in a murine model for AM increased the level of regulatory T cells, which could suppress the cytokine-mediated responses associated with the progression of AM (*Kim et al., 2014*). Although these murine

experimental models were used in various studies on AM, they had several limitations associated with the induction of AD, such as non-specific stimuli owing to hair removal, need for excessive dose of materials (*e.g.*, allergens and chemicals) for sufficient skin application, and unavoidable use of certain mouse strains. To develop an experimental model to addresses the existing limitations, we noted some studies that induced AD using a relatively simple method, such as the application of an allergen on the ears of normal mice (*Choi et al., 2011*; *Choi & Kim, 2014*). Taking a cue from those studies, we attempted to construct a murine experimental model for AM by combining their methods for AD development with the conventional approach for asthma induction without the co-administration of any adjuvant at the sensitization stage, as our preliminary study and other research findings (*Raemdonck et al., 2016*) allowed us to anticipate that asthma was sufficiently induced without an adjuvant. As a result, our findings from the combination model indicated successful establishment of a practical model representing AM-mimicking conditions. Moreover, the end point of AM development in this study indicated that our model showed Th2 asthma endotype, including elevation of IgE, eosinophilic condition, and increased Th2 cytokines (*Lambrecht, Hammad & Fahy, 2019*). These implied that our model was driven by Th2 responses and could be available as a preclinical model to understand common allergic diseases based on type 2 immunity. Taken together, this study demonstrates that our murine experimental model for AM might contribute to improving previous AM models in terms of AD induction by minimizing skin irritation and simplifying allergen application.

Stepwise integrated analyses, including 2-DE, MS, functional annotation, and PPI, revealed that Hspa8 has potential as candidate biomarkers for AM. A number of biomarkers for allergic diseases have been studied using various analyses, such as omics technologies (*Eguiluz-Gracia et al., 2018*; *Zissler et al., 2016*). Although diverse cells and mediators in blood or sputum have been proposed as biomarkers for allergic diseases, AM biomarkers have been poorly investigated, except for several genetic factors. Filaggrin, a well-known predisposing factor for AM, when mutated remains significantly associated with AD and allergen sensitization and increased severity of asthma (*Palmer et al., 2007*; *Thomsen, 2015*). The importance of filaggrin for AM was demonstrated in filaggrin-deficient mice that developed spontaneous dermatitis and pulmonary inflammation (*Saunders et al., 2016*). Several studies have suggested that polymorphisms in the genes encoding thymic stromal lymphopoietin and IL-33 are associated with the risk of AD and asthma (*Harada et al., 2011*; *Margolis et al., 2014*; *Savenije et al., 2014*; *Shimizu et al., 2005*). However, our exploration for candidate biomarkers in this study mainly focused on proteins measurable in biological fluids, which can be obtained more easily. By combining our results from the stepwise functional analyses, starting with exploring proteins differentially expressed in the BALF, we could have determined Hspa8 as a candidate biomarker. Hspa8, also termed heat shock cognate protein 70, belongs to the heat shock protein (HSP) 70 family and plays an important role in protein quality control, such as protein folding and antigen presentation by major histocompatibility complex class II molecules to T cells (*Bonam, Ruff & Muller, 2019*). Hspa8 is also referred to as a major chaperone of the chaperone-mediated autophagy process, which is an intracellular

degradation mechanism, and it acts as a key component that binds to client substrates and delivers them to the lysosome membrane (*Wang & Muller, 2015*). Although several studies have shown that exogenous Hspa8 could suppress lipopolysaccharide-induced inflammation in macrophages and attenuate dysfunction with anti-inflammatory responses in experimental septic shock (*Hsu et al., 2014*; *Sulistyowati et al., 2018*), its function in allergic conditions is still unknown. Hspa8 was constitutively expressed and relatively less expressed during cellular stress, unlike Hsp70, which is otherwise known as a typical stress-inducible protein (*Bonam, Ruff & Muller, 2019*). However, our findings indicated that the expression of Hspa8 might be increased in situations where chronic inflammation persists. Furthermore, the anti-inflammatory response mediated by exogenous Hspa8 is thought to be a key function that contributes to unveiling its role in the immune network and clarifying its association with allergic diseases.

There are several limitations in this study. First, it is controversial that several observations in the asthma-only induced group were less strong; this could be due to the low allergen dose in this study compared to that used in a conventional murine model for asthma. The allergen was administered at a low dose in our model because we assumed that if asthma was strongly induced, it may mask the change of symptoms triggered by AM; further studies are warranted to alleviate this concern. Second, significant associations for Hspa8 were obtained only by statistical analysis, which requires validation by functional or pathophysiological studies. In other respects, it might also be important not to overlook the candidate proteins that are functionally close or more approachable experimentally, even though they showed relatively less significant associations. In this regards, further studies are underway to ascertain if the ferritin light chain, which showed only limited significance in our integrated analyses, might play a certain role in allergic response.

## CONCLUSIONS

In conclusion, we established a murine model for AM that could minimize skin irritation and simplify the application of allergen during AD induction. Based on this improved AM model, we found that Hspa8 had a significant association with AM through stepwise functional analyses. Taken together, our findings provide novel evidence that Hspa8 has potential as a candidate biomarker for AM. Although several studies have refuted the concept of AM and asserted that the prevalence of AM has been overemphasized, it should be recognized that research on AM can provide a new perspective for early prevention, diagnosis, and treatment of allergic diseases (*Yang, Fu & Zhou, 2020*). We expect that our findings will provide better knowledge of experimental models for AM and novel targets for new treatment strategies for allergic diseases.

### Funding

This work was supported by research grants (2016-NI67002-00, 2017-NG67001-00, 2017-NG67001-01, 2017-NG67001-02, 2020-NG-009-00, 2020-NG-009-01) from the Korea

Disease Control and Prevention Agency. The funders had no role in study design, data collection and analysis, decision to publish, or preparation of the manuscript.

### Grant Disclosures
The following grant information was disclosed by the authors:
Korea Disease Control and Prevention Agency: 2016-NI67002-00, 2017-NG67001-00, 2017-NG67001-01, 2017-NG67001-02, 2020-NG-009-00 and 2020-NG-009-01.

### Competing Interests
The authors declare that they have no competing interests.

### Author Contributions
- Kyu-Tae Jeong performed the experiments, analyzed the data, prepared figures and/or tables, authored or reviewed drafts of the paper, and approved the final draft.
- Ji-Hye Do performed the experiments, analyzed the data, prepared figures and/or tables, and approved the final draft.
- Sung-Hee Lee performed the experiments, prepared figures and/or tables, and approved the final draft.
- Jeom-Kyu Lee conceived and designed the experiments, authored or reviewed drafts of the paper, and approved the final draft.
- Woo-Sung Chang conceived and designed the experiments, prepared figures and/or tables, authored or reviewed drafts of the paper, and approved the final draft.

### Animal Ethics
The following information was supplied relating to ethical approvals (*i.e.*, approving body and any reference numbers):

The Institutional Animal Care and Use Committee of the Korea Centers for Disease Control and Prevention approved the study (KCDC-031-16-2A, KCDC-033-17-2A, KCDC-121-17-2A, KCDC-019-19-2A, KCDC-034-20-2A).

### Data Availability
The raw data for Figs. 2–4 is available in the Supplemental Files.

### Supplemental Information
Supplemental information for this article can be found online at http://dx.doi.org/10.7717/peerj.13247#supplemental-information.

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
