# Peer review of "Association of heat shock protein 8 with atopic march in a murine experimental model"

_PeerJ, doi:10.7717/peerj.13247_

## Round 0.1 · original submission · Major Revisions

Please make the corrections and then send it to us with a response letter.

Reviewer 1 ·

Basic reporting

In this study Kyu-Tae et al., propose an improved murine model for atopic march and suggest HSP8 as a candidate biomarker. Authors have applied low-dose of allergen and hence have minimized skin irritation when inducing atopic dermatitis. Using this improved methodology they have assayed IgE level, Th2 response cytokine and lastly have done proteomic analyses to find candidate biomarkers.
The experimental design is methodical and understandable. Manuscript is well-written.

Experimental design

Methodical and well designed

Validity of the findings

Conclusions are well stated and authors have discussed limitations of the study

Additional comments

Authors have done a very good job of writing this manuscript in clear and unambiguous language which is very important to understand the goal of the study. Data provided in the manuscript is also convincing and clean. I also commend authors for discussing the limitations of this study. I have few other comments about this manuscript:
1. In ELISA assays (Fig4), authors did not observe differences in IFN-gamma. To establish that Th response is more of Th2 type and not Th1 type, they can also do ELISA for IL-2 and IL-12, cytokines hallmark of Th1 response.
2. Authors have suggested HSP8 as a candidate biomarker using proteomic data analyses. A simple WB showing expression of HSP8 in the groups compared in the study will further validate this observation. I guess this simple experiment will be under scope of this manuscript.
3. Assays showing TSLP, IL-33 and IL-25 expressions will further validate this improved model for atopic march.
4. Is it possible for authors to provide the fold changes for the proteins depicted in fig 5. It may prove helpful to prospective readers.
5. Lastly, I was just wondering if the list of proteins and (particularly HSP8) the authors have got from proteomics data show modulated expression in response to low-dose of allergen used to induce AD. In future, it will be interesting to see if HSP8 and its interacting partners are still modulated at higher/different doses of allergen. Also, it will be interesting to see if HSP8 and its interacting partners are modulated in other models of atopic march studies. Such studies will help establish HSP8 as a reliable candidate marker for atopic march irrespective of dose or models used for the study. These are just suggestion as a part of peer-review and authors are not needed to respond to this particular comment.

Reviewer 2 ·

Basic reporting

In this manuscript by Jeong et al, the authors establish a modified murine model for studies on the atopic march and this model was confirmed by observation of increased airway inflammation and elevated allergic responses. The authors showed that compared to healthy control group, atopic march induced group exhibited augmented levels of IgE in serum and increased production of Th2 dependent cytokines such as IL-4, IL-5 and IL-13 in BALF. Moreover, using two-dimensional gel electrophoresis and mass spectrometry analysis the authors identified Hspa8 as one of the major differentially expressed proteins in BALF in atopic march induced group. Overall, the study addresses an important finding of Hspa8 as a possible candidate biomarker in atopic march.

The main concerns regarding the manuscript are the inconsistency throughout in describing the treatment duration (weeks) in murine model which creates confusion to the readers, experimental design, and the lack of some important controls. There are numerous minor errors in English expression and punctuation, and general typographic errors, throughout. I suggest the authors should check the entire manuscript carefully and make corrections.

Experimental design

The manuscript could be strengthened by addressing the below listed points.
There is a lot of irregularity in describing treatment duration that creates misunderstanding to the readers:

For example:
Line 27: twice a week for 5 consecutive weeks
Line 90: Once a week
Line 198: once a week for 4 weeks
Line 251: 8 consecutive weeks

Authors should describe the allergen doses and treatment duration in detail where required and keep it consistent throughout the manuscript to make it clear for the reader.

Authors should also represent by arrow AM induced group in Fig1.

Authors used Diff-quick solution to stain different population of immune cells, but it remains unclear how the counting was done and what do they mean by counting 200 cells in each of four different locations. Authors needs to explain in the manuscript how was the quantification performed?

Authors need to briefly describe the acetone precipitation protocol in BALF samples or should cite a reference for the protocol. Moreover, authors need to mention which method was used for protein quantification and what amount of protein were separated by 2-DE.
It was unclear how the treatment of methacholine was done in these experiments? In Fig 3A. authors mentioned that methacholine stimulation was done after 24h of final intranasal challenge, but it remains unclear how it was done for each different mice group, hence I suggest authors should describe in the method section how all the different mice groups (normal control group, AD induced group, AS induced group, and AM induced group) were stimulated with methacholine and at what time point. This needs to be clarified for the reader.

In Fig 3A both airway resistance and dynamic compliance were only significant at higher doses (50mg/ml) of methacholine between AS-induced and AM-induced group. This needs to be explained in the manuscript and authors should clarify the difference between the AS-induced and AM-induced group. While measuring dynamic compliance among groups, why there is so much variability in PBS treated groups. This need to be discussed in the manuscript.

In Fig. 3C how was the scoring performed on H&E-stained sections. Authors need to describe somewhere either in figure legend or in the method section.

Cell Infiltration in lung tissue when compared with NC control group looks very similar to AD-induced group. Any explanation? Authors need to discuss this point in the manuscript.
Did the authors perform an experiment to figure out the different cell types recruited in lung tissue in different mice groups? This would be helpful experiment to study lung pathology in different mice groups and would strengthen the manuscript.

Determining the RNA levels of Hspa8 in the serum or BALF would add more significance to the work. Additionally, comparing the Hspa8 RNA expression in serum/BALF between all four groups would strengthen the conclusions.

Validity of the findings

In Fig 2. Adding statistical significance between week 2 and week 4 of AD-induced group will add more relevance to the data. I suggest keeping consistent with figure legends, replace week 2 and week 4 by Day 8 and Day 22 respectively in the Fig 2B.

In Fig 3B. authors need to perform statistical analysis on different population of cell types between AS-induced group and AM-induced group. Further, authors need to discuss about the decrease observed in monocyte/macrophages and neutrophils in the AM-induced group compared to AS-induced group.

In Fig 4. Statistics is not uniform in all the graphs and hence need to do it in a more consistent manner. There are several places in graphs where statistics analysis needs to be done for example Fig 4 F statistics on AS vs AM induced group is missing. Also, authors should insert non-significant (ns) between the groups where applicable in the graphs.

Additional comments

Minor Concerns:

Line 70 ‘analysis’
Line 98 ‘administered’
Line 108-110 need to improve/reframe the sentence
Line 167 ‘Reference’
Line 213 mention Figure#
Line 220 mention IgE levels were measured in serum
Fig 2A Normal control group received PBS from 0-4 weeks?
Fig 2B Graph scale should be set to ~4000 ng/ml and not to 10000ng/ml since according to raw data provided IgE maximum levels were 3401.856ng/ml
Fig 3 legend insert *p<0.05 versus which group?

Annotated reviews are not available for download in order to protect the identity of reviewers who chose to remain anonymous.

Reviewer 3 ·

Basic reporting

The authors have used professional English throughout the manuscript and the paper was very easy to read. A quick suggestion is that it will be beneficial for the readers to have the full form of the abbreviations spelled out in the figure legends as it makes reading the figures easier.

Experimental design

The authors have developed a schematic experimental protocol and have mentioned a few of the study's shortcomings in the discussion section.
1. The authors are suggested using ANNOVA analysis for Figure 3B which is the appropriate statistical analysis to be used for the particular figure.
2. In figure 3B, apart from eosinophils there were changes observed in macrophages and neutrophils populations in the normal group versus AM group. Authors should discuss this observation in the discussion section as immune cells interact with each other, especially as they are looking at Th1 and Th2 immune responses.
3. Il-10 (Th2) and THF-alpha (Th1) cytokines are well known to be involved with Th2 and Th1 responses respectively. Is there a reason that the authors didn't look at Il-10 and TNF-alpha in BALF for this particular study?
4. In the hub proteins and pathways study, it will be helpful to look at the GO terms and protein enrichment between AM and AD groups. This will provide further detail on proteins being differentially expressed that induce AM in AD groups and compare those proteins with AS group. This will help to understand whether there are any particular groups of proteins that are involved in the progression of AD to allergic asthma to AM.
5. For figure 5, authors should group the proteins (blue spheres) into the cellular pathways/activities they belong in. This will help the readers to understand whether there is any particular cellular pathway that may be involved in the progression of AM from AD.

Validity of the findings

The study showcases a new mice model for AM that addresses the challenges present in the current existing AM models. The authors did a good job of highlighting the shortcomings of the paper. Here are my suggestions to that needs to be addressed in the discussion section:
1. Authors need to discuss the change in the population of neutrophils and macrophages between AD and AM groups along with the eosinophils increased levels. What kind of implications can this have as a whole need to be addressed?

2. Since the GO and PPI analysis were not done between AD and AM groups, we don't know what proteins are involved in the progression. This data will add more value to the paper.

Additional comments

Jeong et.al developed a new mice model to study AM and their analysis showed that heat shock protein 8 is a potential candidate involved in the AM progression. Their mice model addressed some of the challenges present in other published AM models. Overall their study adds value to the AM research and this field.

---

## Round 0.2 · accepted · Accept

Congratulations, you have addressed the shortcomings of the paper in the rebuttal very effectively

Reviewer 1 ·

Basic reporting

In this study, authors propose an improved murine model for atopic march and suggest HSP8 as a candidate biomarker. Using this improved methodology they have assayed IgE level, Th2 response cytokine and lastly have done proteomic analyses to find candidate biomarkers.

Authors have addressed my initial questions in this revised manuscript.

Experimental design

methodical and understandable

Validity of the findings

NO comment

Additional comments

None

Reviewer 2 ·

Basic reporting

Authors appropriately answered all the comments.

Experimental design

Authors appropriately answered all the comments.

Validity of the findings

Authors appropriately answered all the comments.

Additional comments

Authors appropriately answered all the comments.

Reviewer 3 ·

Basic reporting

The authors have written the paper in professional English and the paper is very reader-friendly.

Experimental design

No comments. The authors have satisfactorily answered my questions

Validity of the findings

The study showcases a new mice model for AM that addresses the challenges present in the current existing AM models. The authors have addressed the shortcomings of the paper in their rebuttal very effectively. This study contributes new knowledge to this field.